# Kinetic and Mechanistic Study of Polycarbodiimide Formation from 4,4′-Methylenediphenyl Diisocyanate

**DOI:** 10.3390/ijms26178570

**Published:** 2025-09-03

**Authors:** Marcell D. Csécsi, R. Zsanett Boros, Péter Tóth, László Farkas, Béla Viskolcz

**Affiliations:** 1Institute of Chemistry, University of Miskolc, Miskolc-Egyetemváros, H-3515 Miskolc, Hungary; marcell.daniel.csecsi@uni-miskolc.hu; 2Higher Education and Industrial Cooperation Centre, University of Miskolc, H-3515 Miskolc, Hungary; 3BorsodChem Ltd., Bolyai tér 1, H-3700 Kazincbarcika, Hungary; renata.boros@borsodchem.eu (R.Z.B.); peter.toth45@borsodchem.eu (P.T.); laszlo.farkas@borsodchem.eu (L.F.)

**Keywords:** carbodiimide, MDI, organocatalysis, reaction kinetics, activation energy, oligomerization, transition state theory, DFT

## Abstract

In the polyurethane industry, catalytically generated carbodiimides can modify the properties of isocyanate and, thus, the resulting foams. In this work, a kinetic reaction study was carried out to investigate the formation of a simple, bifunctional carbodiimide from a widely used polyurethane raw material: 4,4′-methylenediphenyl diisocyanate (MDI). The experimental section outlines a catalytic process, using a 3-methyl-1-phenyl-2-phospholene-1-oxide (MPPO) catalyst in ortho-dichlorobenzene (ODCB) solvent, to model industrial circumstances. The reaction produces carbon dioxide, which was observed using gas volumetry at between 50 and 80 °C to obtain kinetic data. A detailed regression analysis with linear and novel nonlinear fits showed that the initial stage of the reaction is second-order, and the temperature dependence of the rate constant is $k(T)=(3.4\pm 3.8)‧10^{6}\cdot e^{\frac{-7192\pm 389}{T}}$. However, the other isocyanate group of MDI reacts with new isocyanate groups and the reaction deviates from the second-order due to oligomer (polycarbodiimide) formation and other side reactions. A linearized Arrhenius equation was used to determine the activation energy of the reaction, which was *E*_a_ = 60.4 ± 3.0 kJ mol^−1^ at the applied temperature range, differing by only 4.6 kJ mol^−1^ from a monoisocyanate-based carbodiimide. In addition to experimental results, computationally derived thermochemical data (from simplified DFT and IRC calculations) were applied in transition state theory (TST) for a comprehensive prediction of rate constants and Arrhenius parameters. As a result, it was found that the activation energy of the carbodiimide bond formation reaction from theoretical and experimental results was independent of the number and position of isocyanate groups, which is consistent with the principle of equal reactivity of functional groups.

## 1. Introduction

Carbodiimides (CDI) with –N=C=N– cumulative double-bond-containing functional groups are well-known compounds in the polymer industry and in biotechnology [1,2,3,4]. In particular, peptide synthesis is common in biological and pharmaceutical applications [2], where some CDIs are widely applied as cross-coupling agents [3]. CDIs can be produced from various compounds including thioureas or isocyanates [1] (pp. 9–36). Isocyanate (–N=C=O, NCO group)-based carbodiimides [4] are the materials of waterproof coatings, adhesives, stabilizer, and most importantly, polyurethanes (PU). The global PU market in 2023 was USD 87.10 billion [5], which makes it a remarkably significant industry. Carbodiimides are highly valued as ageing inhibitors that protect polyester/isocyanate polyurethanes from the effects of heat and moisture [6,7]. The base materials of PUs are polyols and diisocyanates, where methylenediphenyl diisocyanate (MDI) is the most produced isocyanate in the world. MDI has common isomers, mainly 2,2′-, 2,4′-, and 4,4′-MDI depending on the position of NCO groups on the aromatic rings (Figure 1a), while in this paper, 4,4′-MDI is observed (from this point, the abbreviation of MDI refers to the 4,4′-isomer). The melting point of pure monomer MDI is 38.8 °C [8] or 40.41 °C [9]. It is stored between 41 and 45 °C in molten form or below −20 °C (at least below 5 °C) in crystalline form [8].

One specific group of MDI products is manufactured by modifying MDI with carbodiimide bonds and so-called uretonimine compounds. They make the product clear and stable at room temperature in contrast to the pure MDI, facilitating the handling of the material and mixing with co-reactants for PU production [10,11,12]. Uretonimine (often called uretidinone, but the most accurate term is iminouretidinone [13]; abbreviated as UIM in this study) is a four-member ring side-product, which is composed by a [2+2] cycloaddition of the isocyanate group and the formed carbodiimide ([1], Figure 1b), although this can also happen in case of monoisocyanates [13]. This is an equilibrium reaction, where excess monomer MDI can be converted to a 3-functional, 4-membered ring uretonimine, and it also transforms back to initial MDI and CDI [6,13]. The addition of another molecule of MDI or CDI towards UIM can result in the formation of guanidine-like 6-member ring cycloadducts (composed from the initial isocyanate/CDI ratio of 1:2 or 2:1), which accounts for the crosslinking in the material structure of polycarbodiimides [1] (p. 244) [14]. The mentioned side-reactions are shown in Figure 1b.

**Figure 1 ijms-26-08570-f001:**
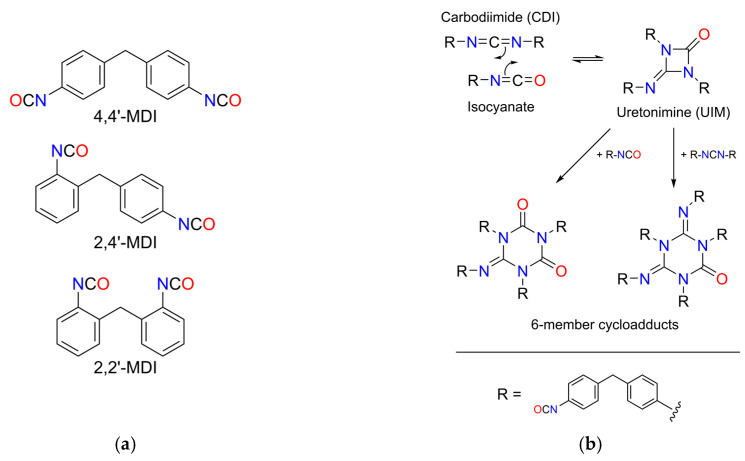
(**a**) Common isomers of MDI (methylenediphenyl diisocyanate). (**b**) Side-reactions, i.e., uretonimine (UIM) and cycloadduct formation, generally and in the case of the R group refers to 4,4′-MDI (based on [1,14]).

Polymeric carbodiimides or polycarbodiimides (PCDI) [1,15,16] are built mainly from diisocyanates with at least two NCO functionals via catalytic step-growth polymerization [17] (pp. 49–51) [18] and polycondensation reaction, eliminating CO_2_ (Figure 2a and Figure 3). Multiple functional groups allow the carbodiimidization reaction further to oligomers. The nomenclature of polycarbodiimides will be PCDI-*n*, where *n* is the monomer number. Likewise, uretonimine products are shortly named as UIM-*n*, depending on the number of MDI substituents around the 4-member ring. PCDI-2, the simplest polycarbodiimide dimer can react catalytically with the remaining MDI or another PCDI-2, yielding PCDI-3 or PCDI-4, respectively. Figure 2b shows the general structure of chain polycarbodiimides.

In this study, the building blocks of industrial, modified MDI products, i.e., polycarbodiimides are presented in the sense of their kinetic formation. According to the literature, acyclic and cyclic phosphine oxides efficiently catalyze CDI conversion from isocyanates even in mild conditions via homogenous catalysis [1,4,19], and 3-methyl-1-phenyl-2-phospholene-1-oxide (MPPO) is one of the most effective catalysts. Furthermore, the industrial application of ortho-dichlorobenzene (ODCB) as a solvent in MDI production [20] makes it suitable and proved to be adequate for examining carbodiimide formation in our previous study [21], where a monoisocyanate, phenyl isocyanate (PhNCO) was observed using computational and experimental methods. It was found that the carbodiimidization reaction insists of two subprocesses (Figure 3) and the kinetic analysis resulted a second-order reaction with the activation energy of *E*_a_ = 55.8 ± 2.1 kJ mol^−1^ [21]. Although, in the case of MDI, the industrially applied temperature range is 80–130 °C [10] or 90–115 °C when using the 1–500 ppm catalyst [11] or 180–280 °C when using the 150–1000 ppb catalyst [22]. In our case, higher temperatures (over 115 °C [11]) would not have been advantageous, due to the excessive formation of side-products (especially uretonimine) and carbon dioxide, which would harden kinetic analysis. Moreover, at higher temperatures, the conversion of MDI into CDI may be autocatalytic in the absence of other catalyst, and it proceeds via an unsymmetrical intermediate cyclic MDI dimer [6,16,23]. Accordingly, the applied catalyst concentration is much higher in our case, and the temperature range was chosen to be lower (50–80 °C). Owing to the impracticability of the detailed instrumental analytical interpretation (e.g., HPLC) of the formed products, gas volumetry was used similarly to follow the course of reaction, in the case of current MDI measurements. This gas burette measurement method was originally described for PhNCO [4] and MDI (Figure 2a, [15]) by Campbell and Monagle, who observed reaction kinetics via directly plotting gas volume data (and not concentrations as in our case). They found that the carbodiimidization reaction follows pseudo-first-order kinetics [4,15] due to the assumption that the catalyst concentration changes in time in the first subprocess. However, regarding the reformation of MPPO in the second subprocess (Figure 3) and higher catalyst concentrations, this theory is inapplicable, and other, detailed kinetic descriptions are desired.

**Figure 3 ijms-26-08570-f003:**
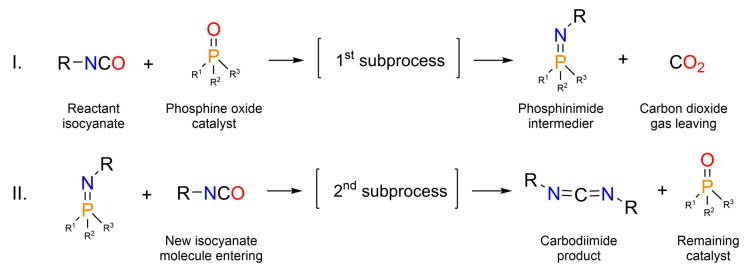
General carbodiimide formation schematically, highlighting the two separate but continuous subprocesses (I and II; [21]), where the R group belongs to the isocyanate and can be both aliphatic and aromatic, and the R^1^, R^2^, and R^3^ groups depend on the type of phosphine oxide catalyst (e.g., tributyl phosphine oxide or 3-methyl-1-phenyl-2-phospholene-l-oxide, i.e., MPPO).

The kinetic evaluation of diagrams requires the consideration of the regression model, in terms of linear and nonlinear functions [24]. Perrin applied the method of least squares to first- and second-order reactions in the case of a “2 A → B” type reaction and concluded that nonlinear curve fitting to the concentration data of reactant is more preferable [25], despite the widespread method of linear fit, using the linearized equation. Due to the heteroscedasticity of data points, the uncertainty of fitting increases as the reaction progresses [25], which can be effectively compensated by weighted linear fits. Moreover, it is better to use nonlinear least-square fit with two variables (*k* and [A]*_t_*_=0_, in second-order kinetics), as fixing one of the two gives a worse fit [25]. The evaluation of kinetic diagrams may include modern rate constant calculation methods [26]; however, the classic method is the analysis of reactions by their rates [27], with the graphical determination of reaction order [28,29]. Besides this, the half-life period method is also applicable [28,29]. Based on these considerations, our datasets were also examined with both nonlinear and linear regression analysis, whilst applying the graphical determination of rate constants.

For a homologous series of different compounds, the most accurate method of estimating thermochemical properties is by group additivity rule [26,30,31]. Group additivity is a well-known theory in physical chemistry and originates from Benson and Buss in 1958 [30]. It proves that thermodynamic properties generally exhibit linear relationships with molecular structures as they are approximately the sum of the thermodynamic contributions of individual substructures within a molecule [26,30]. Furthermore, Flory’s principle of equal reactivity of functional groups is also a similar phenomenon [17,32,33], which proves that at all stages of polymerization the reaction rate and reactivity are independent of the extension of the oligomer chain, only the functional group is decisive (with limitations). These theories were also applied when evaluating results.

The MDI reactions are highly complex due to the two functional groups, thus recognizing that the main kinetic differences between mono- and diisocyanates allows us to deeper understand the chemistry of carbodiimides. The results section includes the detailed kinetic theory of carbodiimidization process, the experimental results, including regression analysis and Arrhenius plots, as well the theoretical investigation of the rate-determining step and the discussion of experimental and theoretical results.

## 2. Results and Discussion

In the results section, the kinetic theory of the complex carbodiimide formation from MDI is described, as is data processing and evaluation, including experimental and theoretical considerations. Diagrams show the properties and determine the kinetic parameters of the supposed reaction.

### 2.1. Reaction Kinetics Background

The reaction chamber initially contains a bifunctional reactant molecule, MDI, where the separate NCO groups react independently with the MPPO catalyst and form carbodiimide. Since the product also has free NCO groups, they also react to different chain-like oligomers, which is schematically depicted in Figure 2b. Furthermore, uretonimine and 6-member ring cycloadduct formation also occurs in addition to carbodiimidization, and thus the amount of reactant MDI is significantly decreased throughout the reaction. In kinetic terms, MDI is involved in parallel reactions, and CDI oligomers take part in consecutive reactions. The separate, most possible gross reactions are shown in Table 1.

Regarding all probable reactions, gross differential kinetic equations can be described to the decrease in MDI (Equation (1)) and increase of CO_2_ (Equation (2)); moreover, to the change in PCDI-2 (Equation (3)). Since uretonimine and 6-ring cycloadduct formation do not generate CO_2_, Equation (2) shows only the MPPO catalyzed carbodiimidization reactions. Reactions that are relevant in terms of the following kinetic evaluation are indicated with bold letters.(1)$$-\frac{d\left[MDI\right]}{dt}=\underset{carbodiimide\;formation}{\underbrace{2k_{1}{\left[MDI\right]}^{2}+k_{2}\left[MDI\right]\left[PCDI-2\right]+k_{3}\left[MDI\right]\left[PCDI-3\right]}}+\underset{\begin{array}{c} \mathrm{uretonimine}\\ \;\mathrm{formation} \end{array}}{\underbrace{k_{U,2}\left[MDI\right]\left[PCDI-2\right]}}+\underset{\begin{array}{c} 6-\mathrm{ring}\;\mathrm{cycloadduct}\\ \mathrm{formation} \end{array}}{\underbrace{k_{C,2}\left[MDI\right]\left[UIM-3\right]}+\cdots }$$(2)$$\frac{d\left[CO_{2}\right]}{dt}=2k_{1}{\left[MDI\right]}^{2}+k_{2}\left[MDI\right]\left[PCDI-2\right]+k_{3}\left[MDI\right]\left[PCDI-3\right]+k_{n}[MDI][PCDI-n]+k_{2,2}{\left[PCDI-2\right]}^{2}+\cdots$$(3)$$\frac{d\left[PCDI-2\right]}{dt}=k_{1}{\left[MDI\right]}^{2}-k_{2}\left[MDI\right]\left[PCDI-2\right]-2k_{2,2}{\left[PCDI-2\right]}^{2}-k_{2,3}\left[PCDI-2\right]\left[PCDI-3\right]-k_{U,2}\left[MDI\right]\left[PCDI-2\right]+\cdots$$
where square brackets show actual concentrations, $k_{1}$, $k_{2}$ etc., are the corresponding rate constants according to Table 1. Since the occurrence probability of the possible reactions and exact rate constants cannot be determined, the gross differential equation to MDI can be simplified if applying the method of initial rates. Oligomer formation has the same apparent rate constants ($k_{1}\approx k_{2}\approx k_{3},\;\ldots$) due to the principle of equal reactivity [32]. Nevertheless, the concentration of oligomers larger than PCDI-2 is negligible at the beginning of the reaction ([PCDI-3] ≈ [PCDI-*n*] ≈ 0), and reaction rate is as follows $r_{1}\gg r_{2},r_{3}$…; thus, the dominant reaction is 2 MDI → PCDI-2 + CO_2_ (Figure 2a) at the initial stage of reactions. This way, simplification can be applied to the overall process, resulting in second-order kinetics, which is in contrast with Campbell’s pseudo-zero-order [15] but in agreement with our similar second-order observations [21]. Results discussed later will also confirm this assumption. This final differential rate equation is shown in Equation (4), where *k* is the observed rate constant, which describes the decrease in MDI and formation of the PCDI-2 product. For plotting MDI decrease on diagram, the well-known integrated version is used (Equation (5)), where $\frac{1}{2{\left[MDI\right]}_{t}}$ is the dependent variable and *t*, time, is the independent variable. The slope of the line gives the desired rate constant *k*, while intercept contains the initial concentration of MDI. Equation (6) shows the predicted half-life of the second-order approximation on the decrease in MDI.(4)$$-\frac{1}{2}\frac{d\left[MDI\right]}{dt}=\frac{d\left[PCDI-2\right]}{dt}=k{\left[MDI\right]}^{2}$$(5)$$\frac{1}{2{\left[MDI\right]}_{t}}=kt+\frac{1}{{2\left[MDI\right]}_{0}}$$(6)t12=12kMDI0

The formation of PCDI-2 is described assuming stoichiometric formation with the formula of Equation (7). Rate constants can be derived from both the decrease in reactant MDI and the increase in product PCDI-2, respectively.(7)$${\left[PCDI-2\right]}_{t}=\frac{{\left[MDI\right]}_{0}}{2}\left(1-\frac{1}{2k{\left[MDI\right]}_{0}t+1}\right)$$

It is important to highlight again that these equations are applicable only at the beginning of the reaction with all the above-mentioned simplifications. The real reaction order is probably a fractional order due to the step-growth polymerization, which is not object of this study.

### 2.2. Kinetic Results

The experimental results section includes diagrams of the primary data (Figure 4a), conversion (Figure 4b), dependence of concentration by time (Figure 5), linearized (Figure 6) and nonlinearized second-order kinetics (Figure 7), their regression analysis and the Arrhenius diagrams (Figure 8). Appendix A shows all sets of raw data and the calculated intermediate data of the measurements.

At first, primary data are processed, which are CO_2_ gas volumes over time. Volume data by temperatures are depicted in a combined gas evolution diagram on Figure 4a. At the beginning of the experiments, the weighed amounts of reactants, i.e., MDI and MPPO, were adjusted to the theoretical maximum of CO_2_, which was calculated to be about 100 cm^3^ due to physical limitations. Nonetheless, due to multiple reactions occurring (oligomerization), after the initial part of the curve, more gas evolutes than the theoretical 100% MDI conversion, as can be seen on Figure 4, especially in the case of 70 and 80 °C measurements. Because of this phenomenon, a notation was applied on the following diagrams, which is an orange dashed line (Figure 4b, Figure 5 and Figure 7), and called it the *limit of applicability*. Above this limit, data points plotted do not show the accurate values (which are unknown with this measuring method), but rather smaller or larger values due to systematic errors of the calculation method. This accompanies the whole calculation process, and the limit for each following diagrams is adjusted to the data intervals of the linear regression. In the following investigation, concentrations of reacted MDI were converted into conversion values to describe the extent of reaction. Conversion vs. time diagram is also plotted below (Figure 4b).

Comparing these diagrams, they pretend to be the same; however, conversion values are calculated from independent initial concentrations, thus the benchmark is slightly different. As a result, conversion diagram is not identical to gas volume diagram, even though in the case of ideally accurate initial weighing and neglected side-reactions, the theoretical maximum of CO_2_ volume would be 100 cm^3^ at 100% MDI conversion. As can be deducted from conversion diagram, the values are overestimated and the two measurements seem to exceed 100%, which occurs due to the mentioned phenomenon and the applied stoichiometric calculation method (Figure 2a, Equation (4)). The assumption is that more MDI is consumed and reacted in the later stage of the reaction than the initial amount, [MDI]_0_, and every mole of CO_2_ comes from the reaction of 2 moles of MDI, which is only valid within the limit of applicability, at the initial stage. Specifically, it seems like that the 60 °C experiment converges to 100%; however, data points are also over this limit, and thus those values are overestimated and are actually much less than 100%. Moreover, the 80 °C measurement is supposed to be the closest to full conversion in the observed temperature range.

The most significant observations during evaluation were performed via calculating concentrations. The derived concentration vs. time plots of the decrease in MDI (Figure 5a) and increase in PCDI-2 (Figure 5b) are depicted below.

Regarding the overall data points, reactant (MDI) concentration decreases, while main product (PCDI-2) increases over time. Comparing these diagrams, the ordinate scale is exactly the half in the case of PCDI-2 (Figure 5b) when compared to MDI (Figure 5a), which consequently indicates that the maximum amount of product cannot exceed the half of the initial MDI concentration, according to Figure 2a and Equation (7). On the other hand, the limit of applicability is highlighted again, owing to the unreliability of further data points; for example, the 70 and 80 °C points in Figure 5a tend to continue below the zero concentration, which is impossible.

Both linearized and nonlinearized fits were performed on the sections of concentration vs. time plots to implement the appropriate analysis of regression. The rate constants are obtained first by plotting the second-order linearized formula via Equation (5) and fitting a line to the initial section (Figure 6), applying the method of initial rates and the method of least squares. Linearized kinetic results are summarized in Table 2, including the coefficient of determination (COD, *R*^2^, *R*-square); furthermore, theoretical half-lives were also calculated from the decrease in MDI via Equation (6).

**Table 2 ijms-26-08570-t002:** Line fitting parameters of the linearized kinetic diagram. The corresponding symbol colours are the same as in Figure 6. Error bars are predicted according to regression software [34].

Temperature (*T*)	Slope, Rate Constant (*k*) (10^−4^ mol^−1^ dm^3^ s^−1^)	Intercept (mol^−1^ dm^3^)	Coefficient of Determination (*R*^2^)	Calculated, Theoretical Half-Life (s)
━ 50 °C	7.26 ± 0.059	0.9334 ± 0.0022	0.9991	1360 ± 11
━ 60 °C	14.4 ± 0.319	0.9140 ± 0.0095	0.9937	715 ± 15
━ 70 °C	30.7 ± 1.680	0.8724 ± 0.0284	0.9795	329 ± 18
━ 80 °C	42.9 ± 2.632	0.8515 ± 0.0353	0.9816	240 ± 15

As shown in Table 2, the initial parts of the curves yield well-fit straight lines (*R*^2^ > 0.979), and thus the assumed second-order kinetics are proven to be applicable to this section of the reaction. Data show that the reaction rate is six times bigger at 80 °C than at 40 °C, and in addition, error intervals only indicate the slight uncertainty error of line fits. Since the initial concentrations of MDI were about 0.5 mol dm^−3^ for all measurements, as a result, the derived intercepts on the diagram (Figure 6) are also roughly the same (Equation (5)). Half-life analysis resulted in values decreasing as temperature increased, and interestingly, they are they are mostly close to the end point of linear fits or, in other words, concentrations are halved at the end of the dominant PCDI-2 formation reaction. Although calculations do not provide the exact time when MDI is the half of the initial amount (due to the applied second-order approximation), but suggests a loose relation between the concentration change and the polymerization process.

Nonlinear regressions are also performed to investigate whether rate constants obtained this way can validate linear regression results, whilst the error of linearization also disappears. For the decrease in MDI concentrations, nonlinear fits are found in Figure 7a. According to Equation (7), a short comparison was carried out on the product formation; thus, the fits for PCDI-2 are also shown in Figure 7b. It should be emphasized that the data intervals of these fits exactly correspond to the linear regressions (Figure 6) and these kinetic results (Table 3) are not used in the further Arrhenius analysis.

**Table 3 ijms-26-08570-t003:** Nonlinear regression parameters in the case of fitting interval are the same as those applied to the linear regressions and in the case of the full dataset being fitted. Derived parameters were the rate constants with error bars, which are calculated by regression software [34]. The parameter [MDI]_0_ of the fitting equations was chosen to be the fixed initial concentrations for each measurement, not the regression parameter. MDI—4,4′-methylenediphenyl diisocyanate, PCDI-2—polycarbodiimide dimer.

Observed Molecule	Number of Regression Points	Temperatures (*T*)	Derived Rate Constants (*k*)(10^−4^ mol^−1^ dm^3^ s^−1^)	Coefficients of Determination (*R*^2^)
MDI	7–15 points, below the limit of applicability	━ 50 °C	6.20 ± 0.18	0.9751
━ 60 °C	11.9 ± 0.36	0.9671
━ 70 °C	23.6 ± 1.39	0.9451
━ 80 °C	32.0 ± 2.38	0.9348
Full dataset	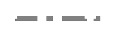 50 °C	6.32 ± 0.24	0.9718
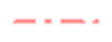 60 °C	16.8 ± 0.90	0.9452
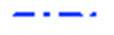 70 °C	34.2 ± 3.60	0.8852
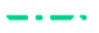 80 °C	56.0 ± 6.83	0.8582
PCDI-2	7–15 points, below the limit of applicability	━ 50 °C	6.52 ± 0.20	0.9739
━ 60 °C	13.4 ± 0.44	0.9626
━ 70 °C	25.9 ± 1.62	0.9407
━ 80 °C	36.8 ± 2.99	0.9273

Comparing nonlinear regression data of PCDI-2 and MDI plots, at lower temperatures, rate constants deviate minimally (0.57–0.65 × 10^−4^ mol^−1^ dm^3^ s^−1^), while at higher temperatures the difference is marginally bigger (2.34–2.58 × 10^−4^ mol^−1^ dm^3^ s^−1^). Nonetheless, in the case of PCDI-2, the reliability of nonlinear regression is limited, as CODs decreasing by increasing temperature. Furthermore, regressions were also performed for the full dataset, which demonstrated that the derived rate constants deviate more from expected values for narrower fitting intervals and CODs show worse fits. The reason for this is the change from second-order approximation above the limit of applicability. This is particularly eveident in the case of the 70 and 80 °C measurements in Figure 7a, where second-order curves start to become horizontal and converge on the abscissa, in contrast to the trend of data points. It was also revealed that the rate constants in the case of linearized fits (Table 2) are slightly bigger than the nonlinearized ones (Figure 7a), and the differences in the corresponding temperatures are negligible, varying between 0.09 and 3.48 × 10^−4^ dm^3^ mol^−1^ s^−1^.

The connection between rate constants and temperature is given by Arrhenius equation. Mostly, the linearized formula is used to plot the correlation between rate constants determined at different temperatures. This ln(*k*) vs. 1/*T* diagram and the linear regression weighted with error bars of rate constants are shown in Figure 8a. The less common, but more accurate nonlinearized Arrhenius diagram is also depicted in Figure 8b, and the equation of fitted Arrhenius curves obtained are also shown in Equations (8) and (9), respectively. (See Appendix A).
Figure 8Arrhenius diagrams made from rate constants obtained from linear regressions (to be consistent with our previous results [21]). (**a**) Linearized diagram (logarithm of rate constants vs. reciprocal temperatures in K) with error bars from Table 2. Equation (8) and the coefficient of determination (*R*^2^) is also presented. (**b**) Nonlinear Arrhenius diagram with nonlinear regression, also regarding error bars. Fitted parameters are shown in Equation (9).
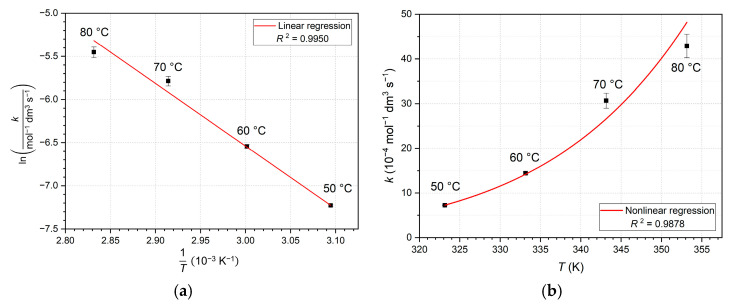

(8)$$\ln k=(-7261\pm 362)\cdot \frac{1}{T}+(15.25\pm 1.12)$$(9)$$k=(3.4\pm 3.8)‧10^{6}\cdot e^{\frac{-7192\pm 389}{T}}$$
where *k* is the temperature dependent rate constant in mol^−1^ dm^3^ s^−1^ and *T* is the temperature in K. Activation energy is calculated from slope of the linear fit on the diagram (Figure 8a) and based on the linearized Arrhenius equation, the energy is given to 60.4 ± 3.0 kJ mol^−1^. It should be noted, that the data are heteroscedastic due to the error considerations, which are mostly significant at 70 and 80 °C (best seen at Figure 8b); thus, this method of determination is subject to greater uncertainty regarding these higher temperature measurements. By analyzing the results, a linearized Arrhenius plot (Figure 8a) predicted the pre-exponential factor with significant errors: $\ln A=15.25\pm 1.12\to A=4.2_{-2.8}^{+8.6}$ × 10^6^ mol^−1^ dm^3^ s^−1^ (from Equation (8); positive error is especially significant), due to exponential dependence, which is more symmetrical in the case of nonlinear plot (Figure 8b): $A=(3.4\pm 3.8)‧10^{6}$ mol^−1^ dm^3^ s^−1^ (from Equation (9)). This phenomenon also reflects on the more favoured use of nonlinear Arrhenius fitting.

Moreover, preliminary calculations were performed in order to investigate whether the observed solution phase reaction is diffusion-controlled or activation-controlled. The value of activation energy is significantly high and the viscosity also does not exert an important influence on rate, considering the formation reaction of the shortest oligomer, PCDI-2. Consequently, the initial part of the reaction is not diffusion-controlled, because results do not show the temperature dependence of the diffusion constant, and the chemical reaction is the determining factor. In the following, the results of the theoretical calculations are presented.

### 2.3. Theoretical Calculations

Computational chemistry calculations were carried out to support the experimental results and obtain preliminary information about the mechanism of polycarbodiimide formation. In our first article about carbodiimides [21], the full mechanism of the CDI formation is discussed, regarding the two subprocesses (Figure 3), complexes, transition states and thermochemical results (Appendix A) calculated by Gaussian, using B3LYP/6-31G(d)[SMD(ODCB)] level of theory. Calculations showed that the rate-determining step is the TS3 transition state in the first subprocess, where CO_2_ separates from the complex and exits from the system (Figure 3) [21]. The driving force here is this leaving molecule, which increases the entropy of the system, thus lowering Gibbs free energy and letting the reaction occur. Current calculations in case of MDI were also performed at the same level of theory, but indicating only the significant step, the initial molecules (MDI, MPPO) and the rate-determining step (TS, which refers to the previously called TS3 [21]). Structures (Figure 9 and Appendix A; Appendix A) were geometrically optimized followed by frequency calculations and intrinsic reaction coordinate (IRC) analysis (Figure 10, Appendix A), which ensured that the results were adequate. The thermochemical evaluation showed that the relative enthalpy of activation of the rate-determining transition state (TS) compared to the initial energy of reactants (MDI + MPPO) was 52.89 kJ mol^−1^ (Appendix A), and the further analysis of the intermediate steps was omitted.

Furthermore, knowing the theoretically determined enthalpy and entropy of activation of the reaction (Appendix A) at different temperatures, transition state theory (TST) can be applied to determine theoretical rate constants and the activation energy, which are comparable with experimental values. The modified Eyring—Polányi equation is shown below (Equation (10)):(10)$$k=\kappa \varepsilon \frac{k_{B}T}{hc^{⦵}}e^{\frac{\Delta S^{‡}}{R}}e^{\frac{-\Delta H^{‡}}{RT}},$$
where *k* is the second-order rate constant (mol dm^−3^ s^−1^), κ is the transmission coefficient (κ = 1), ε is a correction factor we applied, *k*_B_ is the Boltzmann constant, *T* is the absolute temperature (K), *h* is the Planck constant, $c^{⦵}$ is the standard concentration ($c^{⦵}$ = 1 mol dm^−3^), $\Delta S^{‡}$ is the entropy of activation at *T* temperature (J mol^−1^ K^−1^), $\Delta H^{‡}$ is the enthalpy of activation at *T* temperature (J mol^−1^), and *R* is the gas constant. Energies were determined from the 298.15 K standard computational calculations, using the heat capacity of activation ($\Delta C_{V}^{‡}$) on the same temperature range as the experiments, i.e., 50–80 °C. Results showed that the rate constants that were derived from Equation (10) and corrected with heat capacity, were plotted with the corresponding temperatures, yielding a new Arrhenius diagram, and the predicted activation energy was 56.3 kJ mol^−1^ and the pre-exponential factor was 3.80 × 10^3^ mol^−1^ dm^3^ s^−1^ (fitting errors here are negligible since the *k* values are roughly multiples of each other and *R*^2^ = 1.00). However, this determination method is subject to the errors of assuming the ideal TST (κ = 1) and a harmonic oscillator model, as well as a low level of theory and improper entropy calculations. Since the pre-exponential factors obtained from DFT calculations omit the solvent effect, an empirical ε factor was applied as an additional correction to Equation (10) by comparing the experimental results. The corrected pre-exponential factor and the comparison of results are presented in the discussion.

Inspections reveal that TST is also applicable to predicting theoretical activation parameters from experimentally measured rate constants, as well as Arrhenius theory. Although, this case is not discussed in detail, a brief comparison (Appendix A) showed that both the enthalpy and entropy of activation derived from linearized Eyring equation reproduced the theoretical energies (at 298.15 K), using the data of Table 2. Values are $\Delta H^{‡}$ = 57.6 kJ mol^−1^ and $\Delta S^{‡}$ = −127.4 J mol^−1^ K^−1^, respectively.

### 2.4. Discussion of Kinetic Results

The comparison of different experimental regressions and theoretical methods with each other and the bifunctional MDI with the monofunctional phenyl isocyanate (PhNCO) [21] represent a perfect base for comparison. Due to the various results, a combined Arrhenius diagram of different fits (Figure 11), its derived regression parameters (Table 4), and a comparison matrix are also presented (Table 5).

**Table 4 ijms-26-08570-t004:** Line fitting parameters of the combined Arrhenius diagram. The corresponding symbol colours are the same as in Figure 11. Error bars are predicted according to regression software [34]. TST—transition state theory.

Source of Rate Constants	Activation Energy, Derived from Slope (kJ mol^−1^)	Pre-Exponential Factor Derived from Intercept (mol^−1^ dm^3^ s^−1^)	Coefficient of Determination (*R*^2^)
━ Linearized fit	60.4 ± 3.0	$4.2_{-2.8}^{+8.6}$ × 10^6^	0.9950
━ Nonlinearized fit (full dataset)	74.6 ± 6.3	$7.5_{-6.7}^{+67.7}$ × 10^8^	0.9858
━ Nonlinearized fit (initial part)	55.9 ± 4.0	$6.8_{-5.2}^{+22.0}$ × 10^5^	0.9900
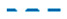 Theoretical, TST	56.3	8.8 × 10^5^ *	1.0000

* After empirical correction in TST, ε=232.8.

Figure 11 compares kinetic data from linearized and nonlinearized regressions and theoretical results too. The high deviation of nonlinearized fit for the full dataset in all parameters indicates that the reaction is not second-order throughout the full progress, as oligomerization changes it. Conversely, the rate constants derived from the nonlinear fits to the initial section of the kinetic curve show better correlation with the ones from the linear regression and the theoretical ones, demonstrating that the second-order approximation is true at the beginning of the reaction, which also applies to the CODs. The data suggest that slight uncertainty in the intercepts plays major role in the value of pre-exponential factors derived from them, which appears in asymmetric error bars, but mainly in the upper deviation. Theoretical calculations include the *k* constants obtained from Eyring equation, followed by empirical correction, where the ε factor was considered to be ε=232.8 to ensure consistent comparison. As a result, this pre-exponential factor reproduces the result of nonlinear initial fit in the same order of magnitude. The following comparison is made in order to draw conclusions using literature data.

**Table 5 ijms-26-08570-t005:** Comparison matrix of summarized results, including experimental data from linear regressions for both MDI (4,4′-methylenediphenyl diisocyanate) and our previously published PhNCO (phenyl isocyanate) [21]. The comparison of the theoretically obtained barrier of enthalpy of activation at 298.15 K is also shown. TS3 refers to the same rate-determining step for MDI (TS in Figure 9 and Figure 10) that was examined for PhNCO.

		PhNCO (“mono”)	MDI (“di”)
Experimental measurements	Activation energy, *E*_a_ (kJ mol^−1^)	55.8 ± 2.1 [21]	60.4 ± 3.0
	Pre-exponential factor, *A* (mol^−1^ dm^3^ s^−1^)	(4.2 ± 2.7) × 10^5^ [21]	$4.2_{-2.8}^{+8.6}$ × 10^6^
Theoretical calculations	R–TS3 enthalpy of activation, $\Delta H^{‡}$ (kJ mol^−1^)	52.87 [21]	52.89

Table 5 reflects on the strongly similar results of the activation energies (average difference is 4.6 kJ mol^−1^ in relation to PhNCO), as well as the pre-exponential factors with similar orders of magnitude (difference is explained by the fact that the initial molar concentration is approx. half in the case of MDI, in contrast to the case of PhNCO). This observation shows that the molecular interactions take place similarly according to collision theory and TST [29], which reflects the previously discussed principle of equal reactivity and the effect of group additivity, as the results are independent of the type of observed isocyanate and also not regarding the number of NCO functionalities. Moreover, the more popular $\frac{1}{{\left[MDI\right]}_{t}}=k\cdot t+\frac{1}{{\left[MDI\right]}_{0}}$ second-order kinetic formula could also be used during the analysis, instead of the applied, more accurate Equation (5). This formula neglects the exact stoichiometry according to the gross equation (Figure 2a); however, the Arrhenius parameters calculated this way would remain the same.

Highlighting theoretical calculations, the TS enthalpy of activation for MDI (52.89 kJ mol^−1^) was compared with the case of PhNCO, which was 52.87 kJ mol^−1^ [21]. The two values have a good correlation, as the difference is only 0.02 kJ mol^−1^, which is negligible; thus the theoretical calculations also confirm that the carbodiimidization reaction is independent of the type of isocyanate. This phenomenon is similar to the experimentally compared results (Table 5), and the activation energy is also in good agreement with the theoretical one. Furthermore, if we apply a temperature correction to the obtained activation enthalpy, according to the equation $E_{a}=\Delta H^{‡}+RT$ [29] (p. 114), then activation energy is finally given as 55.37 kJ mol^−1^. This is even closer to the experimental value, 60.4 ± 3.0 kJ mol^−1^ (difference is only 5.0 kJ mol^−1^), than the enthalpy, even though theoretical calculations were performed at 298.15 K. Based on this latter observation, it can be seen that the theoretical calculations support and confirm our developed experimental method.

## 3. Materials and Methods

Experiments were carried out using MDI (Ongronat^®^ 3000, BorsodChem, Kazincbarcika, Hungary) as a reactant, ODCB (distilled) as a solvent, and MPPO (INTATRADE Chemicals, Muldestausee, Germany) as a catalyst, which were measured with an OHAUS^®^ Explorer analytical balance (0.0001 g resolution). The non-jacketed gas burette (max. 100 cm^3^, 0.2 cm^3^ resolution) and level-bottle were filled with confining liquid, which is an acidic and saturated sodium sulphate solution to reduce the absorption of CO_2_. It was coloured with methylene blue indicator. Burette and level-bottle were connected by a plastic tube. Temperature was measured with a digital thermometer. Kinetic data analysis and data visualization were performed using a simple algorythm written in Spyder Version 6.0, and using a plotting software, namely OriginPro 2018 [34]. The fitting uncertainty of rate constants were taken into consideration with instrumental weighting by error bars when editing Arrhenius diagrams.

Theoretical calculations were executed using the Gaussian 16 programme package [35] on a high-performance computer (HPC), and molecules were visualized in GaussView 6.0.16 [36]. The structures of the complex and transition state were optimized by applying density functional theory (DFT) in addition to the B3LYP hybrid functional [37] and 6-31G(d) basis set. The SMD implicit solvent model [38] was used for ODCB (*ɛ*_r_ = 9.9949) in order to simulate experimental conditions.

### 3.1. Experimental Apparatus

The experimental apparatus consisted of a reaction vessel (F) in which the reaction mixture took place. The reaction solution contained 0.2 wt% MPPO and 10 wt% MDI, which was diluted enough not to allow side-reactions. First, a catalyst solution was made from approx. 0.04 g (0.0405 ± 0.0003 g) of MPPO and was diluted to 18 g with ODCB using analytical balance. The mass was measured and was introduced into the pre-thermostated vessel. Above it, a reflux Graham condenser (D) was connected for cooling, in case evaporation was found to be a determining factor and to keep the gas at a roughly constant temperature (26.5–28.3 °C). To enhance accuracy, the applied gas temperature was also measured and used in the calculations, although recording was not continuous. Regarding the low melting point of MDI (38.8 °C), the predefined temperatures were 50, 60, 70, and 80 °C. In addition, the handling of MDI requires signigicant attention, as, due to its physical properties (i.e., crystallizing at room temperature), all tools must be preheated to always keep the material in molten state for addition and measurement. At the same time, a magnetic stirrer (G) was switched on and a Pasteur pipette was preheated in a drying oven. Using the pipette, ca. 2 g MDI (2.0013 ± 0.0157 g) was measured and added instantly to the vessel; the weight was measured back with the balance (during the removal and the addition of the material, the pipette also cools down enough to produce tiny crystals; thus, a little more, about 2.06 g MDI, should be removed from the MDI vessel). Simultaneously, a stopwatch was started, and a three-way stopcock (H) was set from the position of outside atmosphere (I) to the gas burette (J), which was filled with the confining liquid. To provide constant pressure in the system, a compensative level-bottle (K) was used with the burette, the position of which was changed continuously using a laboratory scissor jack (L) as more carbon dioxide evoluted. A detailed, informative figure is presented here (Figure 12).

**Figure 12 ijms-26-08570-f012:**
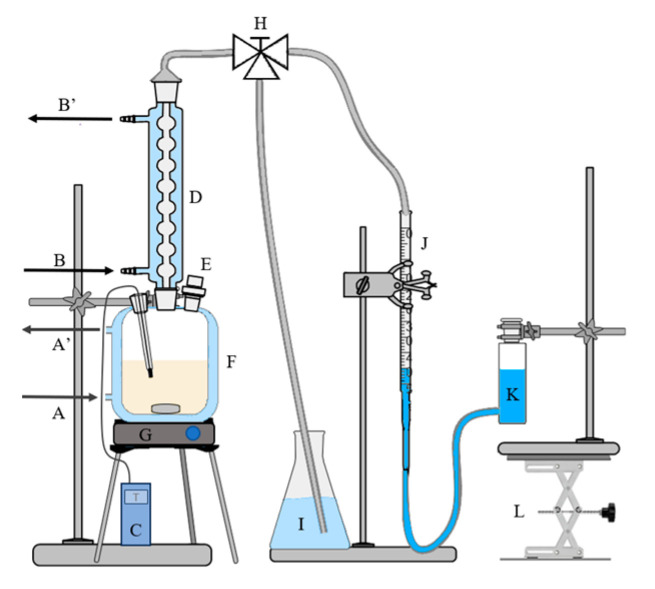
The experimental apparatus. A, B—thermostat inlets; A’, B’—thermostat outlets; C—thermometer; D—reflux condenser; E—inlet for reactant input; F—reaction vessel; G—magnetic stirrer; H—three-way stopcock; I—atmospheric equilibrium flask; J—gas burette; K—level-bottle; L—laboratory scissor. An additional thermometer was placed to the top of condenser to measure gas temperature. Source: adapted from [21].

### 3.2. Calculations, Evaluation Process

To obtain kinetic data, gas volume values over time ($V_{g,t}$) were handled as main primary data. Gas volumes were applied to determine moles and, indirectly, the concentration and conversion of MDI. Figure 13 shows a short summary of the calculation process. Actual amounts of carbon dioxide ($n_{g,t}$ (mol)) were calculated using general gas law with constant pressure (*p* = 101,325 Pa, neglecting the vapour pressure of confining liquid) and corrected with the monitored gas temperature ($T_{g,t}$). It was proven to be sufficient as van der Waals equations do not provide very accurate results.

According to Figure 13, to calculate actual isocyanate concentration ${\left[MDI\right]}_{t}$, MDI moles ($n_{MDI,t}$) and the actual volume of solution ($V_{S,t}$) are required. The values of current, reacted MDI moles can be obtained from gas moles using stoichiometry, as shown in Figure 2a, $n_{MDI,t}=2n_{g,t}$; this is also the case for the product PCDI-2: $n_{PCDI-2,t}=n_{g,t}$. The volume of the reaction solution at time *t* is estimated using a correction from the mass decrease via the elimination of the gas ($m_{S,t}=m_{S,0}-m_{g,t}$) and the constant density (*ρ*), $V_{S,t}=\frac{m_{S,0}-{(n}_{g,t}\cdot M_{CO_{2}})}{\rho }$, where $m_{S,0}$ is the initially measured mass of solution (g), $n_{g,t}$ are the previously calculated current overall CO_2_ moles (mol), and $M_{CO_{2}}$ is the molar mass of carbon dioxide (g mol^−1^). The density of solution was measured via a model solution composed of 10 wt% MDI diluted with ODCB at 50 °C and was quantified as *ρ* = 1.277 g cm^−3^. An observable error is the increasing number of oligomers during the reaction, which increases average molecular mass, viscosity, and even density. This effect was not considered in our calculations but is negligible during the initial second-order kinetic phase. Another potential error is the temperature dependence of density, which was also not considered. The actual MDI concentration is given using the formula ${\left[MDI\right]}_{t}=\frac{{n_{MDI,0}-n}_{MDI,t}}{V_{S,t}}$ and the conversion $X_{t}=\frac{n_{MDI,t}}{n_{MDI,0}}$.

### 3.3. Methodology and Data Analysis

Regression analysis was performed initially using the linearized kinetic plots, where the COD was selected to be 0.9800, which was found to be reliable for carrying out consistent data analysis. Each temperature included 2–3 measurements to enhance reproducibility, and the least squares method and Python code were applied to determine fitted lines at most points of a dataset that fulfil the *R*-square criterion (e.g., 15 points for 50 °C and 7 points for 80 °C measurements, see Figure 6).

Regarding the exact input concentrations of MDI and MPPO, the percentage of MDI, wt% (MDI), and the mass ratio of MPPO/MDI were calculated in all experiments carried out and compared with each other. However, in the sense of obtaining rate constants, the determining effect is only the MPPO/MDI ratio in contrast to the percentage of MDI. All presented datasets meet the conditions.

Some corrections were applied to ensure appropriate calculation process. Regarding the measurement of gas temperature, temperature-dependent and independent formulas were also tried. It was revealed that the applied temperature-dependent formula results in insignificant differences in gas moles compared to the temperature-independent formula. Furthermore, as the gas temperature increase is at most 2 °C over the course of a measurement, according to Gay-Lussac’s law, the volume growth is barely 0.6% (approx. 0.4 cm^3^, close to the resolution of burette); thus, the measurement of the gas temperature is not particularly significant. The correction of the vapour pressure of solvents on the gas volume data was not considered in these calculations as it was considered in a previous study [4]. Moreover, the effects of slightly changing the volumes of the solution were also analyzed. The values differed from 15.43 to 15.59 cm^3^ throughout the measurement, and thus the total differences between the initial and final volumes varied between 0.09 cm^3^ and 0.15 cm^3^ at the 50 and 80 °C measurements, respectively. Volumes decreased over time, and this very small growth (0.09 → 0.15) was a result of the ever-increasing amount of CO_2_ as the temperature increased. Nevertheless, this is at most only a 0.96% change in the volume, so simplifications could also be applied here, and this factor may also be negligible.

## 4. Conclusions

In this study, the kinetic analysis of polycarbodiimide formation from MDI was discussed. MDI contains two NCO functionals, which react with each other in the presence of a phospholene oxide catalyst, MPPO, if heated. The multi-step condensation reaction produces carbon dioxide, which was measured with a conventional gas burette over time. The kinetic theory includes the rate equations of the formation of different PCDI oligomers, uretonimines, and other cycloadducts. The rate-determining reaction is the reaction of two monomer MDI molecules, which can be described with second-order kinetics. Several measurement cycles were performed experimentally in the temperature range between 50 and 80 °C. In data evaluation, CO_2_ volumes were converted to concentrations, and the systematic linear fits were applied to obtain rate constants and Arrhenius plots with error bars. Through regression analysis, it was found that the beginning of the linearized second-order diagram is linear below the so-called limit of applicability, whilst the latter points deviate, which shows the oligomer and side-product formation. This is also validated by fitting to the full dataset and by comparing linear and nonlinear regressions to the reactant decrease and product formation. By applying the method of initial rates, the rate constants and final Arrhenius parameters showed that the experimental activation energy of PCDI-2 formation is 60.4 ± 3.0 kJ mol^−1^, according to the linearized Arrhenius diagram. This energy value was compared using computational chemistry calculations. The formation of PCDI-2 from MDI was calculated, and the enthalpy of activation of the rate-determining transition state was revealed to be 52.9 kJ mol^−1^ at the B3LYP level of theory, where CO_2_ molecules exit the complex. Brief IRC analysis validated this local maximum via PES. Comparing this value with our previously calculated monofunctional phenyl isocyanate, they exhibited good agreement, and consistent with the principle of the equal reactivity of functional groups and the independence of the substituents, as observed in both carbodiimide- (from PhNCO) and polycarbodiimide formation (from MDI). Nonetheless, applying the Eyring equation and TST ensured a comprehensive comparison between theoretical and experimental calculations.

In our future research, we plan to examine the rheokinetic characteristics of the resulting polycarbodiimide gel and also carry out a detailed computational study of the forming oligomers and uretonimine species. The presented kinetic and mechanistic analysis contributes to our understanding of isocyanate–carbodiimide as well as to polyurethane chemistry from an atomistic view.

## Figures and Tables

**Figure 2 ijms-26-08570-f002:**
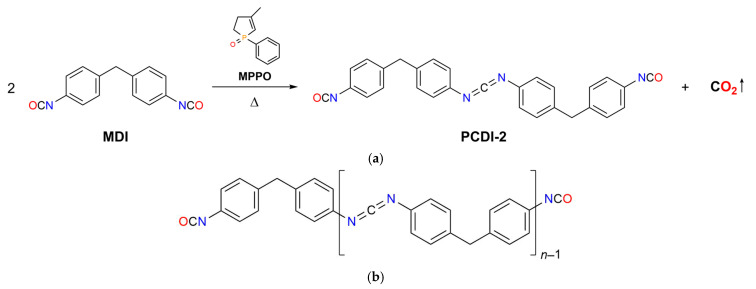
(**a**) Reaction equation of the formation of the simplest, polycarbodiimide dimer (PCDI-2) from MDI (4,4′-methylenediphenyl diisocyanate), indicating the need of MPPO (3-methyl-1-phenyl-2-phospholene-l-oxide) catalyst and heat, while carbon dioxide gas eliminates. (**b**) The expected general chain polymer product (PCDI) of carbodiimidization reaction, initiated from MDI, where *n* indicates the number of monomer MDI molecules. *n* = 1 case refers to the initial MDI itself.

**Figure 4 ijms-26-08570-f004:**
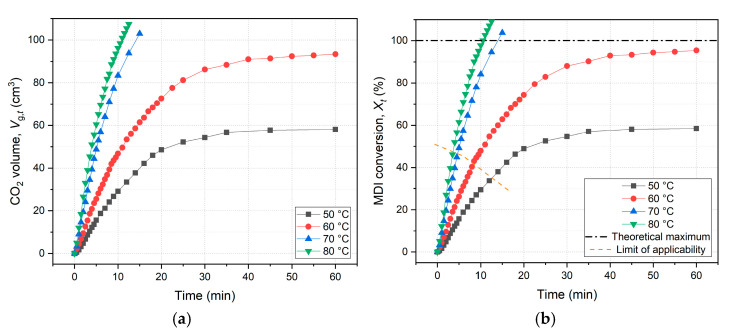
(**a**) Gas volume and (**b**) conversion data by temperatures. Above the limit of applicability, data points are inaccurate, as second-order approximation becomes inadequate. The slight difference between the two diagrams is due to the minor deviations in the initial concentrations of the individual measurements. MDI—4,4′-methylenediphenyl diisocyanate.

**Figure 5 ijms-26-08570-f005:**
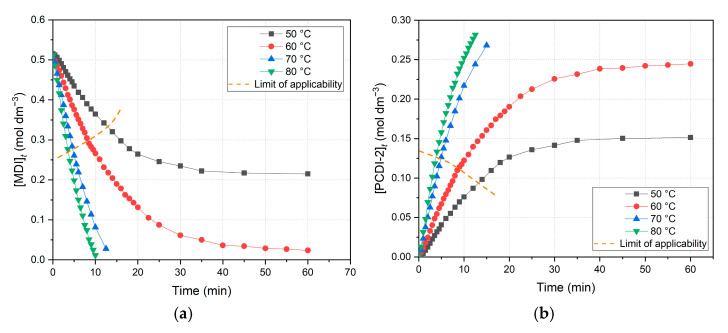
Full experimental kinetic diagrams at all temperatures (**a**) on the decrease in MDI (4,4′-methylenediphenyl diisocyanate) reactant concentrations and (**b**) the increase PCDI-2 (polycarbodiimide dimer) product concentrations, calculated using the stoichiometric method. Above the limit of applicability, data points are inaccurate, as second-order approximation becomes inadequate.

**Figure 6 ijms-26-08570-f006:**
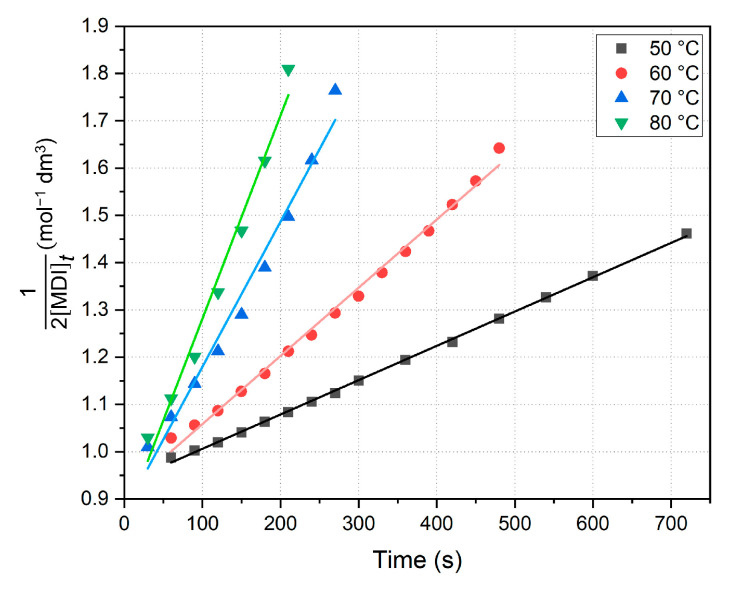
Summarized kinetic diagram by linearized second-order formula. Fitted parameters are shown below in Table 2. MDI—4,4′-methylenediphenyl diisocyanate.

**Figure 7 ijms-26-08570-f007:**
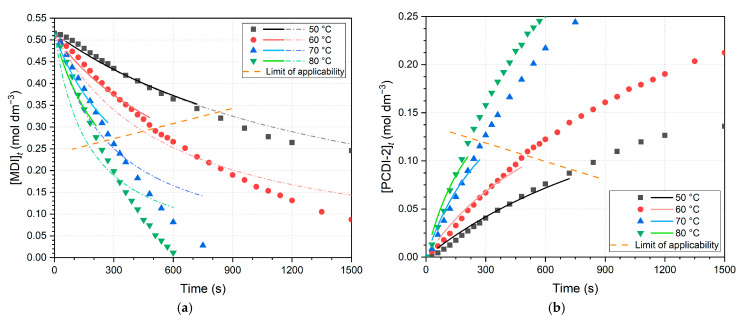
Nonlinearized curves fit to the initial part of the reaction, within the limit of applicability. This is shown in the case of (**a**) MDI (4,4′-methylenediphenyl diisocyanate), where dash dot curves symbolize regressions for the full dataset (not only in the range of depicted points), and in the case of (**b**) PCDI-2 (polycarbodiimide dimer). Above the limit of applicability, oligomerization dominates and second-order approximation becomes inadequate.

**Figure 9 ijms-26-08570-f009:**
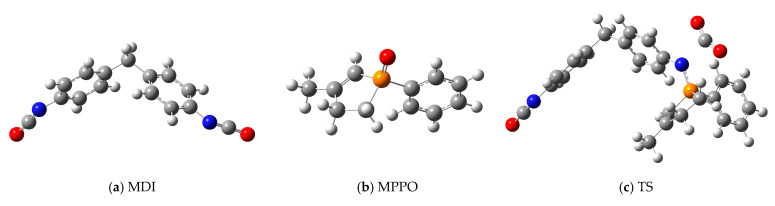
Optimized structures visualized in GaussView programme, calculated at the B3LYP/6-31G(d)[SMD(ODCB)] level of theory, 1 atm and 298.15 K. (**a**) MDI—4,4′-methylenediphenyl diisocyanate; (**b**) MPPO—3-methyl-1-phenyl-2-phospholene-1-oxide; (**c**) TS—transition state of the rate-determining step.

**Figure 10 ijms-26-08570-f010:**
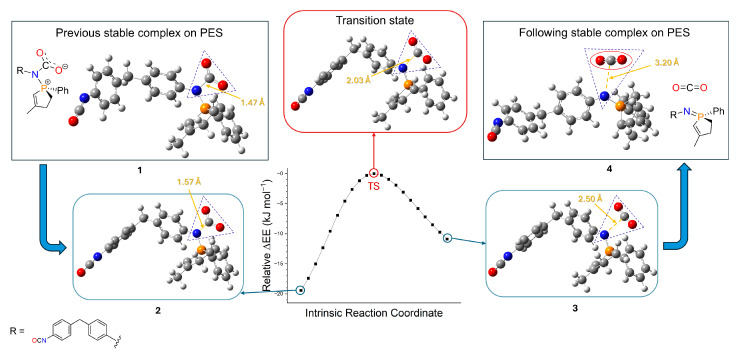
Intrinsic reaction coordinate (IRC) analysis on the potential energy surface (PES) in the 1^st^ subprocess of the reaction. Data show relative electronic energies (EEs) to the rate-determining transition state (in red rounded rectangle), with 10 points each direction. The edging points (**2** and **3** in blue rounded rectangles) are close to the corresponding stable complexes (local minima; **1** and **4**), and the structures indicate the separation step of the CO_2_ molecule (red oval). The four significant atoms (purple triangles) as well as the bond lengths (yellow) are also shown for comparison. In the schematic structures, R group refers to 4,4′-methylenediphenyl diisocyanate (MDI) and Ph refers to the phenyl group. ΔEE (**1**) = −30.8 kJ mol^−1^; ΔEE (**2**) = −19.4 kJ mol^−1^; ΔEE (**3**) = −10.9 kJ mol^−1^; ΔEE (**4**) = −22.1 kJ mol^−1^.

**Figure 11 ijms-26-08570-f011:**
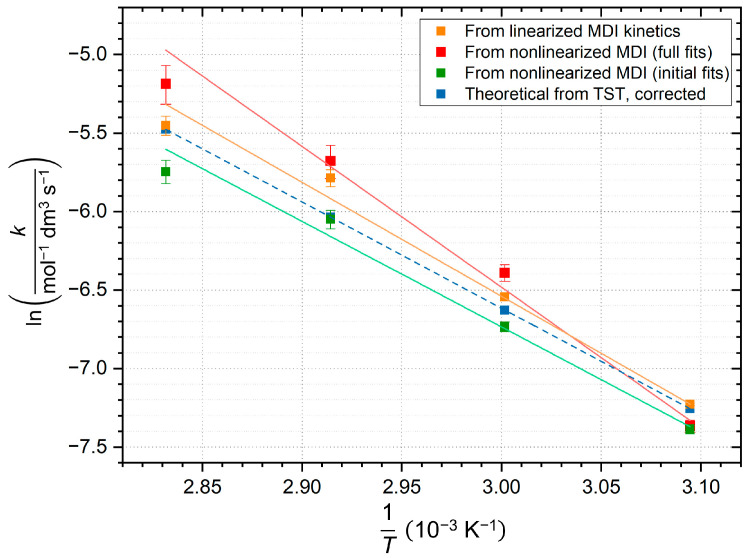
Combined, linearized Arrhenius diagram, indicating the different rate constant data for the decrease of MDI (4,4′-methylenediphenyl diisocyanate) concentration, weighted with error bars, derived from linear fits (Figure 6 and Figure 8a, Table 2, orange), from nonlinear fits at the initial part (Figure 7a, Table 3, green), from nonlinear fits for the full dataset (Figure 7a, Table 3, red), and from data derived theoretically from transition state theory (TST), followed by empirical correction (blue, dash line).

**Figure 13 ijms-26-08570-f013:**
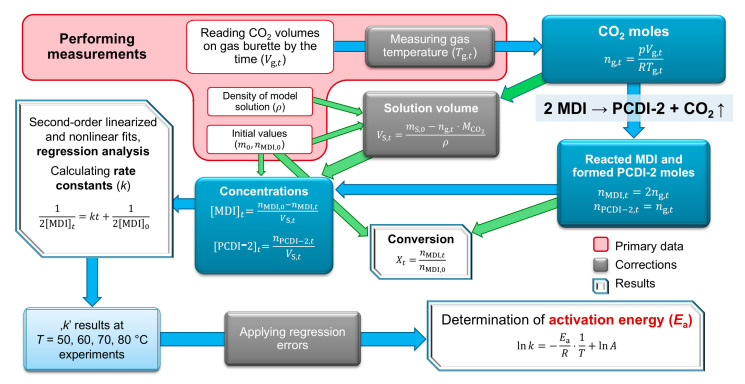
Calculation pathway from measurement to activation energy determination in terms of the substitution in SI units. Blue arrows indicate the main calculation route and green arrows are for supporting calculations, assuming that all the CO_2_ comes only from the monomer MDI. The model solution consists of 10 wt% MDI (4,4′-methylenediphenyl diisocyanate) and 90 wt% ortho-dichlorobenzene (ODCB). Explanations of the formulas are provided in the text below. PCDI-2—polycarbodiimide dimer.

**Table 1 ijms-26-08570-t001:** The most probable reactions occurring in the reaction vessel. No statistical data on probability are available due to the complexity of the system. MDI—4,4′-methylenediphenyl diisocyanate; PCDI-*n*—polycarbodiimide oligomer containing number *n* MDI monomers; MPPO—3-methyl-1-phenyl-2-phospholene-1-oxide; UIM—uretonimine.

Added Segment/Product	Reactions	Reaction Type	Reaction Rate
+ MDI	2 MDI → PCDI-2 + CO_2_ *	Catalytic processes with MPPO, forming CO_2_	$r_{1}=k_{1}{\left[MDI\right]}^{2}$
	MDI + PCDI-2 → PCDI-3 + CO_2_		$r_{2}=k_{2}[MDI][PCDI2]$
	MDI + PCDI-3 → PCDI-4 + CO_2_		$r_{3}=k_{3}[MDI][PCDI3]$
	MDI + PCDI-*n* → PCDI-(*n*+1) + CO_2_		$r_{n}=k_{n}[MDI][PCDIn]$
+ PCDI-*n*	2 PCDI-2 → PCDI-4 + CO_2_		$r_{2,2}=k_{2,2}{\left[PCDI2\right]}^{2}$
	PCDI-2 + PCDI-3 → PCDI-5 + CO_2_		$r_{2,3}=k_{2,3}[PCDI2][PCDI3]$
Uretonimine	MDI + PCDI-2 ⇌ UIM-3 MDI + PCDI-3 ⇌ UIM-4	[2+2] cyclo-additions, without forming CO_2_	$r_{U,2}=k_{U,2}[MDI][PCDI2]$ $r_{U,3}=k_{U,3}[MDI][PCDI3]$
6-member ring cycloadducts	MDI + UIM-3 → P1 PCDI-2 + UIM-3 → P2		$r_{C,2}=k_{C,2}[MDI][UIM3]$ $r_{C,3}=k_{C,3}[PCDI2][UIM3]$

* The most likely reaction to occur at the initial stage of the process.

## Data Availability

The raw data presented in this study are available upon reasonable request from the corresponding author.

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
