# Peer review of "Kinetic and Mechanistic Study of Polycarbodiimide Formation from 4,4′-Methylenediphenyl Diisocyanate"

_ijms, 2025, doi:10.3390/ijms26178570_

Round 1

Reviewer 1 Report

Comments and Suggestions for Authors

Regrettably, the manuscript of Csécsi et al. does not contain enough new achievements to justify its publication in this journal. Therefore, it must be rejected.

Main Reasons for Manuscript Rejection:

[1] Originality: Notwithstanding the kinetics investigation is well-described and explained, it lacks novelty and presents nothing new to the existing literature.

[2] Contribution: The molecules used are well-known in the literature, as demonstrated in the introduction. The proposed results, while well-explored, also do not offer any new insights. This lack of originality extends to the experimental methodology for the kinetics analysis and the computational approach.

[3] DFT Analysis: The analysis using DFT (Density Functional Theory) calculations is simplistic and underdeveloped. The molecular model used to describe the mechanism was overly simplified and fails to explore the diversity of the experimental findings.

[4] Methodological Choice: The B3LYP method, while historically significant in computational chemistry, has been extensively investigated. The literature has already demonstrated its ineffectiveness for the specific type of analysis the authors are performing.

[5] Basis Set: The use of the 6-31G(d) basis set compromises the methodology's accuracy in describing electronic energy and thermodynamic properties, rendering it inappropriate for the investigation's purpose.

[6] Transition State structure (TS): The description of the TS, which was the only one observed by the authors, was not presented appropriately.

[7] Introduction: The introduction is excessively long. Its eight paragraphs make it difficult to discern the main purpose of the study. Many paragraphs are redundant and could be more concise to make them more objective

[8] References: The references are outdated and very old, failing to demonstrate the relevance and currency of the research. Many of the references are more than six decades old.

Author Response

Please, find attached.

Reviewer 2 Report

Comments and Suggestions for Authors

Please note that the detailed review report is provided in the attached file.

Author Response

Please, find attached.

Reviewer 3 Report

Comments and Suggestions for Authors

This manuscript reports a kinetic and computational investigation of polycarbodiimide formation from 4,4′-methylenediphenyl diisocyanate (MDI) catalyzed by 3-methyl-1-phenyl-2-phospholene 1-oxide (MPPO) in orthodichlorobenzene (ODCB), monitored via volumetric COâ‚‚ evolution. Rate constants are determined at multiple temperatures using linear and nonlinear regression, and activation energy is calculated via Arrhenius analysis. DFT calculations are used to support a proposed rate-limiting transition state.

While the data are technically sound and methodologically consistent, the study closely mirrors the authors' prior work on phenyl isocyanate (Ref 23), employing the same catalyst, solvent, and analytical approach. No new chemical transformations, mechanistic pathways, or analytical techniques are introduced. The manuscript does not clearly articulate its novelty, significance, or broader relevance…particularly in the title, abstract, and conclusion. As such, the work may be better suited to a specialized journal in catalysis or polymer chemistry rather than IJMS, which prioritizes molecular insights with broader biological or interdisciplinary impact.

Questions/Suggestions:

  • What new mechanistic or kinetic insights does this MDI-based study provide beyond Ref 23 (phenyl isocyanate)?
  • How does this justify a standalone publication using the same catalyst, solvent, and methodology?
  • The current title (“prelude of polymerization”) is ambiguous and grammatically flawed. Consider revising to clearly reflect the study’s focus (e.g., Kinetic and Mechanistic Study of Polycarbodiimide Formation from MDI Catalyzed by MPPO).
  • The abstract and conclusion largely restate procedures. Please revise to articulate the study’s key findings, implications, and relevance.
  • Was product characterization (e.g., NMR, IR, MS) performed to confirm the structure and purity of the polycarbodiimides? If not, please discuss the limitations of relying solely on COâ‚‚ evolution.
  • The assumption that higher oligomers (e.g., PCDI-3, -4) are negligible in the early stage needs validation. Can you provide experimental data or a mass balance?
  • Were side reactions (e.g., uretonimine formation) or intermediates evaluated, either experimentally or through DFT?
  • The DFT analysis focuses only on a single transition state. Have entropic contributions, competing pathways, or multiple TSs been considered?
  • The concept of a “limit of applicability” for second-order kinetics is interesting but underexplained. A visual schematic or annotated plot could help readers interpret this threshold more clearly.
  • If this study is intended as groundwork for polymerization, what are the implications for MDI-based polymer design or catalyst optimization? How does this kinetic analysis inform future synthetic directions?

Author Response

Please, find attached.

Round 2

Reviewer 1 Report

Comments and Suggestions for Authors

Regrettably, the manuscript does not contain enough new achievements to justify its publication in this journal. Therefore, it must be rejected.

Characterization of the Reaction Results:

The apparatus developed for CO2 detection is interesting, but that is all. It measures only CO2, which is the sole product that can be detected. This is like a single-result sensor: it only detects CO2 elimination. However, several important questions are left unanswered: What happens beyond the CO2 elimination? The answer to this question: We cannot know. Therefore, the title itself is invalidated, since the only mechanism that can be experimentally detected is CO2 production.

DFT Results:

The presentation of the DFT results has improved, but there has been no improvement beyond that. One point that stands out is the fact that the authors did not explore other potential pathways, which is understandable given the experimental issues I mentioned previously. However, with computational methods, they could at least explore and prove that this is the most likely mechanism, rather than simply presenting it as the only one that can be compared to the experimental results.

The computational methodology remains the same. The authors could have used more appropriate methods, many of which have well-established accuracy in the literature. If the problem was an inability to reproduce the transition state structure with another method, there are numerous composite methods (even using other DFT methods for single-point calculations) to refine the electronic energy. One example from a huge range of options is r2SCAN-3c. Alternatively, composite methods based on post-Hartree-Fock methods, such as G3MP2, which the group has used in the past, would be preferable. Using the B3LYP/6-31G(d) method is not justified for a system of this size.

Critical Point in an Overview of the Study:

One point catches my attention: The authors used a computational method with low accuracy (a well-known limitation of this method), but in this case, it apparently achieved accuracy with the experimental results. The research, however, seems to end here. What should have been done is to explore the validity of this result for confirmation purposes using more accurate methods, and perhaps even with multiple methodologies if the group's computational infrastructure cannot extend beyond DFT. Furthermore, as I mentioned, other approaches should also have been explored to demonstrate that the proposed mechanism is the only possible one under the given pressure and temperature conditions.

The same applies to the experimental part. Given the limitations of the apparatus used to characterize the reaction kinetics, the authors should have used other resources to further explore the reaction.

Reviewer 2 Report

Comments and Suggestions for Authors

No commets

Reviewer 3 Report

Comments and Suggestions for Authors

The authors have substantially revised their manuscript and addressed nearly all of the concerns raised in the first round. The revised version is clearer, more complete, and places the study in better context. The supporting information has been strengthened, and the cover letter provides helpful clarifications.